# Rainfall regimes, their transitions, and long-term changes during Indian summer monsoon

Bhupendra A. Raut<sup>1</sup>, Aditi Deshpande<sup>2</sup>, Devyani Kamble<sup>2</sup>, Sandip Ingle<sup>3</sup>, Parmeshwar Naik<sup>4</sup>, Shwetal Walde<sup>5</sup>, P. Pradeep Kumar<sup>2</sup>, and Purnendranath Sen<sup>2</sup>

**Correspondence:** A. Deshpande (aditid84@gmail.com)

Abstract. We present a diagnostic framework and accompanying dataset of daily rainfall regimes during the Indian Summer Monsoon (ISM) for June–September 1961–2018. Using high-resolution (0.25°) daily rainfall and unsupervised k-means clustering, eleven objectively defined spatial rainfall patterns were identified and linked with characteristic low-level winds, sea-level pressure and moisture fields, separating different modes of active and break phases. The dataset provides (i) centroid rainfall patterns of each regime and (ii) daily cluster IDs, enabling reconstruction of the full temporal sequence of rainfall regimes and calculation of transition probabilities between states. Transition analysis confirms that break phases are the most persistent while monsoon depressions are more transient, mirroring observed synoptic life cycles. A decomposition of rainfall change between 1961–1989 and 1990–2018 shows that drying in Northeast India ( $\sim$ 9%) is driven by fewer break periods with northeast-focused rainfall, whereas Gangetic Plain drying ( $\sim$ 7%) is linked to both intensity and frequency changes. This regime-based approach provides a powerful diagnostic tool to examine synoptic drivers, long-term changes in rainfall intensity and frequency, regime transition dynamics, and to evaluate model representation of ISM variability, teleconnections and trend attribution.

## 1 Introduction

Indian Summer Monsoon (ISM), typically occurring from June to September, is a primary source of water for the agricultural sector, industrial activities and urban regions, and with its variability has a strong impact on the country's economy and foodgrain production (Gadgil and Gadgil, 2006). The ISM is a planetary scale seasonal weather pattern characterized by changes in atmospheric pressure and seasonal reversal of continental-scale surface winds that bring consistent rainfall to the Indian subcontinent (Ramage, 1971; Rao, 1976).

<sup>&</sup>lt;sup>1</sup>Environmental Science Division, Argonne National Laboratory, Lemont, IL-60439, USA

<sup>&</sup>lt;sup>2</sup>Department of Atmospheric and Space Sciences, Savitribai Phule Pune University, Pune-411007, India

<sup>&</sup>lt;sup>3</sup>Indian Institute of Tropical Meteorology, Ministry of Earth Sciences, Govt. of India, Pune-411008, India

<sup>&</sup>lt;sup>4</sup>Skymet Weather Service Pvt Ltd, Department of Agrometeorology, Agricultural College, Pune-411005, India

<sup>&</sup>lt;sup>5</sup>Symbiosis Institute of Geo-Informatics, Symbiosis International (Deemed University), Pune-411016, India

20



The primary processes attributed to this system is the differential heating of the Indian subcontinental landmass that causes moist air to rise over land, creating a low pressure and driving moist winds from the southern Indian Ocean (near Mascerene High) towards the Indian landmass. However, the singular role of this land-sea temperature gradient has been disputed since the early 1920s (Simpson, 1921). Additionally, the influence of the Intertropical Convergence Zone (ITCZ) in governing the seasonal wind and rainfall shifts, particularly its northward progression that sets up the Asian summer monsoon circulation and determines rainfall distribution, has been highlighted by Sikka and Gadgil (1980). These strong moist southwesterly winds are integral to sustaining monsoon.

The pre-monsoon phase typically occurs from March to May and is characterized by hot and dry conditions over the Indian subcontinent. The monsoon onset phase typically occurs in June and is characterized by the arrival of the monsoon winds (Southwesterly) and the beginning of heavy rainfall which is observed first over Kerala coast (Ananthakrishnan and Soman, 1988). The seasonal mean rainfall is modulated by active/break monsoon phases which typically occur once the monsoon is established over the entire subcontinent, and is characterized by the strongest/weak winds and heaviest/low rainfall (Raghavan, 1973; Murakami, 1976; Goswami and Mohan, 2001). The withdrawal monsoon phase typically occurs during September and is characterized by the weakening of the south-westerlies and the gradual decrease in rainfall (Rajeevan et al., 2006). The strength of the meridional pressure gradient is governed by the low pressure over north Indian and Pakistan region and high pressure over Mascarene Islands, named Mascarene High (Sikka and Gadgil, 1980). This gradient is also important in determining the strength of the cross-equatorial jet. In addition, monsoon depressions, monsoon trough, mid-tropospheric cyclones heavily influence the distribution and variability of rainfall across the subcontinent (Rao, 1961).

The monsoon trough is a low pressure system that forms during the summer monsoon season over the north India, typically extending east-west and dips in to the head Bay of Bengal. The monsoon trough acts as a channel for the flow of moist air from the ocean onto the land, leading to the formation of clouds and heavy rainfall, with rainfall maxima observed to the south of the trough (Rao, 1976; Webster et al., 1998). The position of the monsoon trough affects the distribution of rainfall over the Indian subcontinent and is also an important indicator of changes in active to break phases of monsoon. During active phases the trough is located further south than usual giving more rainfall over the southern and central parts of the subcontinent. During the break phase, trough moves at the foothills of Himalaya, giving more rainfall near the foothills of Himalaya and north-eastern parts of the subcontinent (Rao, 1976).

Monsoon depressions are low-pressure systems that form and propagate along the monsoon trough when the surface waters of the Indian Ocean is anomalously warm. These monsoon depressions then move towards the Indian subcontinent causing heavy and prolonged rainfall along its path(Rajeevan et al., 2010). They also change the direction and strength of the monsoon winds, which affect the distribution of rainfall over the region.

Mid-tropospheric cyclones, exhibiting maximum intensity around the 500 mb level during the summer are characterized by a distinct thermal structure, with a warm anomaly situated above the cyclone and a cold anomaly below (Krishnamurti and Hawkins, 1970). Such thermal configurations induce rising motions west of the cyclone and sinking motions to its east (Goswami et al., 1980). Their presence modulate monsoon rainfall, as they govern the ascent of moist air, leading to significant convective rainfall events (Choudhury et al., 2018).



In addition, orographic effects play a critical role in geographic distribution of rainfall patterns (Gadgil, 1977). The Western Ghats, in particular, exhibit a pronounced orographic effect on rainfall during the southwest monsoon (Sarker, 1966; Phadtare et al., 2022). The Himalayas, not only act as a barrier for cold mid-lattitude winds but also increase the ascending motion and rainfall over the Gangetic plains during active phases and foothills of Himalayan region during break phases, with orography playing a significant role (Raghavan, 1973). Thus the complex interplay between orographic features and the prevailing atmospheric conditions, particularly the of lower tropospheric westerlies shapes the rainfall distribution.

Intraseasonal variations in monsoon rainfall is largely determined by the movement and interactions of the synoptic weather systems. However, these synoptic systems are modulated by large scale variability modes that show distinct signatures at intraseasonal, interannual and long term timescales in monsoon intensity and distribution (Gadgil, 2003; Webster et al., 1998). The Madden–Julian Oscillation (MJO) (?) and the El Niño–Southern Oscillation (ENSO) (Kumar et al., 1999) are the primary large-scale modes modulating ISM rainfall intensity and variability, along with the Indian Ocean Dipole (IOD), Pacific Decadal Oscillation (PDO), and the Arctic Oscillation (AO) (Saji et al., 1999; Ihara et al., 2007; Thompson and Wallace, 1998; Deshpande et al., 2017, 2014). IOD modulates the impact of ENSO on the ISM (Ashok et al., 2007). Positive IOD is characterized by the warming of the west equatorial Indian Ocean and anomalous cooling of east equatorial Indian Ocean, which enhances rainfall over the Indian subcontinent. On the other hand, the negative IOD can decrease rainfall over the subcontinent (Ashok et al., 2001; Deshpande et al., 2014). The relationship between the Arctic Oscillation (AO) and the Indian monsoon is least explored in comparison to other climate teleconnections like ENSO. However, Gong et al. (2001) showed the linkage between the Arctic Oscillation and the East Asian winter monsoon and Bamzai and Shukla (1999) discussed the linkage between Eurasian snow cover and the Indian summer monsoon. In presence of the multi-scale variability and intricate teleconnections, the modulation of Indian monsoon rainfall on daily scale is predominantly dictated the daily synoptic circulations and mesoscale weather systems.

The Indian Summer Monsoon (ISM) has displayed significant temporal variations. An analysis of APHRODITE data from 1951 to 2007 showed an increase inter-annual rainfall variations across India (Duncan et al., 2013). An increasing trend in drought severity is also observed in recent decades in south India and the Indo-Gangetic plains (Mallya et al., 2016). Bajrang et al. (2023) also observed a decrease in monsoonal precipitation extremes over Central India from 2005 to 2020. Over a longer term (1901 to 2022) individual spell contributions varied, however the total rainfall contributions remained consistent (Subrahmanyam et al., 2023). This is possibly due to multidecadal rainfall trend reversals (Chakra et al., 2023). It is evident that the quantification of changes in both frequency and intensity of rainfall over different region is necessary to gain a deeper understanding of evolving monsoon rainfall. Straus (2022) clustered 5-day anomaly of 850 hPa horizontal winds from the ERA-Interim reanalysis to study active-break cycles and intra-seasonal oscillations. (Falga and Wang, 2022) found significant increases in extreme events in nine regions from 1901-2020 using clustering, linking them to urbanization and other climatological factors.

In addition to changes in rainfall patterns on synoptic, intraseaonsal and interannual scale, there is evidence of increasing extreme rainfall events (Ghosh et al., 2012; Roxy and Chaithra, 2018; Goswami et al., 2006) which translates in to increased floods and drought risks occurring in the same season. The moisture-holding capacity of the atmosphere increases at roughly

7% per degree Celsius (Clausius–Clapeyron relation), suggesting a propensity for more intense rainfall events with increasing temperatures (Trenberth, 2011; Held and Soden, 2006). Additionally, climate projections indicate possible shifts in the timing of monsoon onset and retreat and an amplification of intraseasonal variability, all of which could adversely impact agriculture and water management (Roxy and Chaithra, 2018; Turner and Annamalai, 2012). Future projections show that rainfall is estimated to increase while monsoon circulation will weaken in the twenty first century (Kulkarni et al., 2020).

This study uses a clustering method with the aim to (1) characterize the synoptic-scale rainfall patterns of the Indian monsoon by linking them to known atmospheric circulations, (2) to understand their sequential evolution during the monsoon season, and (3) to determine how changes in these patterns contribute to the spatially heterogeneous trends in monsoon rainfall. By providing a clustering analysis framework, this work offers a method for improve evaluation of model simulations. Our clustering dataset will be useful in future studies examining monsoon teleconnections and model biases, following the precedents of regime-based analysis of (e.g., Raut et al., 2014, 2017).

## 2 Data and Methods

#### 2.1 IMD data




Indian Meteorological Department (IMD) has released high-resolution  $0.25^{\circ} \times 0.25^{\circ}$  daily rainfall data since 1901 over the Indian subcontinent (Pai et al., 2014; Srivastava et al., 2009). The dataset has been created using rain gauge data from 6955 rain gauge stations from the archives, with varying periods of availability. Three sets of data from IMD archives combined with Aphrodite data were used over the Indian subcontinent (Yatagai et al., 2012). The observatories are divided between IMD observatory stations, hydro meteorology observatories, agromet observatories. A detailed description of all four datasets and the methodology is given in Pai et al., 2014. We have used daily means of  $0.25^{\circ} \times 0.25^{\circ}$  rainfall data from the India Meteorological Department (IMD) over June-September from 1961 to 2018 to compute k-means clusters as per the methodology used in (Raut et al., 2014). Additional linear trends analysis, frequency, and intensity analysis were also conducted using this dataset.

## 2.2 NCEP-NCAR data

NCEP reanalysis data is produced by the National Centers for Environmental Prediction (NCEP) that provides a comprehensive record of global weather and climate conditions. The dataset is produced by combining observations from a variety of sources, including weather stations, and satellites, with a numerical weather prediction model. This allows researchers to create a consistent and complete record of weather and climate conditions, going back several decades.

In this study, we used 20°S-35°N to 40°E-120°E domain over the Indian region for studying synoptic environment. Daily anomalies for mean sea level pressure (MSLP), wind (at 850mb, 700mb & 500mb), and specific humidity (at the surface) were calculated using the National Center of Environmental Prediction-National Center for Atmospheric Research (NCEP-NCAR) reanalysis-1 (https://psl.noaa.gov/data/gridded/data.ncep.reanalysis.html) dataset (Kalnay et al., 1996). NCEP-NCAR have produced a 40-year record of global reanalyses of atmospheric fields keeping quality control and data assimilation system



unchanged over the reanalysis period 1948 to present. The database has been enhanced with many sources of observations not available in real time for operations.

## 2.3 Trends Analysis

To quantify long-term changes in monsoon rainfall, we perform linear trend analysis on regional average rainfall. We also divided the 58-year record into two periods (1961–1989 and 1990–2018), and computed the difference in mean rainfall between these two periods as an estimate of multi-decadal change. Additionally, for selected regions we computed the linear trend of JJAS total rainfall over 1961–2018. The statistical significance of trends is evaluated using the Mann–Kendall test (not shown in detail for brevity, but significance at the 95% level is noted where relevant). All trend are calculated on unsmoothed data, however we applied a 5-point Gaussian-weighted running mean for visualization in Figure 3. Linear trends were not always appropriate, therefore, we used a LOESS (Locally Weighted Scatterplot Smoothing Cleveland, 1979) curve to highlight multi-decadal fluctuations in cluster frequencies over time in Figure 9.

## 2.4 Clustering Rainfall

## 2.4.1 k-means Clustering Algorithm

k-means is an unsupervised clustering algorithm that divides a dataset into a specified number of clusters (k) based on the similarity of the data points (Anderberg, 2014). It takes an n-dimensional data set comprising of m vectors and works by first selecting k initial centroids and then assigning each data point to the cluster whose centroid is closest to them in Euclidean space. The centroids are then updated to be the mean of the data points in their respective clusters, and the data points are reassigned to the new clusters based on the updated centroids. This process is repeated until the centroids converge and the assignments of data points to clusters become stable. The goal of k-means is to minimize the cumulative sum of the Euclidean distances across all points in the clusters, as illustrated in Eq. 1.

$$J = \sum_{i=1}^{n} \sum_{j=1}^{k} w_{ij} \|x_i - \mu_j\|^2$$
(1)

However, one limitation of the k-means algorithm is its dependence on the user specifying the precise number of clusters. Moreover, its efficacy can be influenced by the initial conditions for certain data sets and k values.

## 2.4.2 Finding Rainfall Threshold

Seasonal rainfall in Western Australia occurs in bursts due to frontal passage or easterly dips, causing approximately 38% dry days, and close to 48% light rain days. Hence, the dominant cluster could contains approximately 60% members with all dry days and some light rain days. To mitigate this, Raut et. al. (2014) differentiated dry days from rainy days using a threshold and clustered the rainy days to capture more variability within the rainfall patterns. In contrast, during the Indian monsoons, the distribution of daily mean area rainfall across the entire domain, as depicted in Figure 1, follows a Gaussian distribution

**Figure 1.** The frequency distribution of daily area mean rainfall for the study region is normally distributed with approximately 1% of days with no rain in the domain during JJAS. Therefore, all the days were clustered.

with negligible occurrences of dry days. Even during the break monsoon periods, the foothills and southern India often feature intense localized rain, while the rest of the subcontinent remains relatively dry. Therefore, we retained all days in our analysis and obtained the clusters of comparable sizes.

## 2.4.3 Finding Optimal k

While the determination of k can sometimes be guided by prior knowledge, theoretical insights, or objective methods, in situations where no pre-existing knowledge about the probable number of clusters in the dataset is available, the various statistical metrics are used. We utilized the Elbow method as described by Bholowalia and Kumar (2014), as our recent studies established it as a robust method (Raut et al., 2021; Jackson et al., 2023). By executing the algorithm with incrementally increasing k from 3 to 20, we plotted the changes in the intra-cluster sum of squared Euclidean distances, known as the Within Cluster Sum of Squares (WCSS), and the inter-cluster sum of squared distances, termed the Between Cluster Sum of Squares (BCSS), as functions of k (See Fig. 2). A noticeable break in the downward/upward trends of WCSS/BCSS at k = 11 suggested that 11 clusters are optimum. To mitigate potential issues arising from the algorithm's sensitivity to the initial seeding as mentioned above, we assessed the stability of our clustering by repeatedly initializing the k-means algorithm and observing any shifts in cluster assignments. As this process is computationally expensive for a large dataset, we used five iterations. Throughout five iterations, fewer than 0.1% of any cluster's members altered their cluster affiliation, indicating the presence of

**Figure 2.** In the k-means algorithm, the user has to provide k. We employed commonly used elbow criteria for two metrics as a function of k a) change in WCSS and b) change in BCSS. The optimal k is the smallest number of compact and separable clusters.

stable clusters when k = 11. The kmeans function in R programming language, was employed on June-September 1961-2018 rainfall data, using the algorithm of Hartigan and Wong (1979), with initial seed equal to double the number of clusters  $(2 \times k)$  and a limit of 100 iterations.

# 2.5 Transition Probabilities

With each day labeled by a cluster, we treated the sequence as a discrete-time Markov chain of "weather states" to understand the probabilities of transitioning from one cluster to another. The general formula for the transition probability from state i to state j in a Markov chain is denoted by  $P_{ij}$ .

$$P_{ij} = \frac{N_{ij}}{\sum_{k=1}^{n} N_{ik}} \tag{2}$$

where  $N_{ij}$  is the number of transitions from cluster i on day d to cluster j on day d+1, and the denominator is the total number of days in cluster i (minus one if the last day of a season is cluster i, since it has no next day within the season, however, given the long period analyzed and continuity across seasons year-to-year, this edge effect is negligible).

# 2.6 Change analysis

The change in total rainfall across two periods, R1 and R2, can be parsed into two components 1. attributed to changes in intensity and 2. attributed to changes in frequency. To compute these changes for each cluster, we utilized the method described in Catto et al. (2012) and Raut et al. (2014).

$$R_i = N_i \Delta P_i + P_i \Delta N_i + \Delta N_i \Delta P_i \tag{3}$$

$$R = \sum_{i=1}^{k} R_i \tag{4}$$

Here,  $N_i$  and  $P_i$  denote the frequency of occurrence and intensity of the  $i^{th}$  rainfall cluster during period 1, respectively. Meanwhile,  $\Delta N_i$  and  $\Delta P_i$  represent the changes in frequency and intensity for the  $i^{th}$  cluster from period 1 to period 2.

The first and second terms in Eq. 3 can be understood as the modifications in rainfall attributed to changes in intensity and frequency, respectively. The third term is a higher-order Taylor series expansion component representing shifts in rainfall due to concurrent changes in intensity and frequency. Given that this term is a product of the two changes, it's usually small. However, if the dataset exhibits high variability, the third term could potentially exceed the changes attributed to intensity (first term), frequency (second term), or both. In such scenarios, any change term surpassed by the third term is deemed non-significant.

#### 3 Results


## 190 3.1 Spatial Analysis of Trends

Before analyzing the clusters, we first describe the spatial distribution and recent trends of monsoon rainfall over India to provide context. Figure 3 depicts the average summer monsoon rainfall in India during the June-September period (JJAS), highlighting the Western Ghats, Central India, and Northeast India regions which typically receive the most rainfall. Nevertheless, this distribution varies on multiple spatio temporal scales. To explore the long-term variations in seasonal rainfall, the study period of 1961-2018 was split into two sub-periods: 1961-1989 (denoted as R1) and 1990-2018 (denoted as R2). Figure 3b and 3c illustrate the changes in rainfall between these periods, revealing a decline in recent rainfall over most parts of the subcontinent, except over arid and semiarid regions of western India. Specifically, prominent changes in widespread rainfall were detected in the regions along the monsoon trough. There is an increase in rainfall over the Thar and Kutch region (T&K), while the Indo-Gangetic plains (IGP) and Northeast India (NEI) witnessed a reduction. Figure 3c presents the rainfall change

**Figure 3.** a. Daily mean area rainfall b. Change in daily mean rainfall between two periods 1961-1989 (R1) and 1990-2018 (R2). C. Same as b but in percentages. The locally weighted smooth curves (blue lines) and linear trends (black line) in annual accumulations of rainfall in the regions bounded by the boxes shown in c. for d Thar and Kutch, e. Indo-Gangetic plains, and f. Northeast Indian region.

(R2-R1) in percentage terms, whereas Figures 3d-f show the linear trend in average rainfall for the aforementioned regions. Over the 58-year span, the T&K region (Figure 3d) experienced a consistent rise in rainfall with a large decadal fluctuations, in contrast to the steady decreases in the Indo-Gangetic plains (Figure 3e) and Northeast India (Figure 3f). Although recent studies have shown an increase in extreme rainfall events, (Goswami et al., 2006), but a weakening in Indian circulation (Turner and Annamalai, 2012), the data from Figure 3 suggests that these shifts are not consistent throughout the subcontinent. In this study we focused on the monsoon trough and the areas influenced by monsoon depressions, as these systems are the dominant drivers of daily and intraseasonal variability and critically shape the long-term distribution and trends of rainfall.







## 3.2 Characteristics of Rainfall Regimes

Applying the k-means algorithm to the daily rainfall data yielded 11 distinct clusters, whose mean rainfall patterns are shown in Figure 4. Associated circulation patterns shown with sea level pressure and 850 hPa wind anomalies in Figure 5, while 850 hPa geopotential height and precipitable water anomalies are shown in Figure 6. Although, cluster transitions are discussed in Section 3.3, we will refer to the Figure 7 in the current section when necessary. The active monsoon clusters have significantly higher and widespread rainfall, cyclonic anomaly in wind and pressure chart and higher moisture content (clusters 1, 4, 6, 7, 9 and 11) and the break monsoon clusters are with a lower rainfall, anticyclonic anomaly and lower moisture content in the subcontinent (clusters 2, 3, 5, 8, 10). These clusters can be grouped based on rainfall patterns, wind and specific humidity anomalies (at 1000mb), and mean sea level pressure, as follow.

- 1. **Prolong break** (**P-B**) [Cluster 3]: Cluster 3 is the most frequent cluster (27.6% of days) and synoptic pattern corresponds to the typical break-monsoon condition. Rainfall is largely suppressed across the entire subcontinent except for light rain along the Himalayan foothills and parts of the southeast peninsula. The widespread dryness is evident from the 






- whereas Cluster 10 is almost exclusively northeast-focused. Transitions among these clusters are common (Section 3.3), indicating they often succeed each other in time as part of a break spell.
  - 3. Monsoon depressions, active phase (MD-A) [4, 11, 9]: These clusters represent active monsoon conditions dominated by the presence of monsoon low-pressure systems and/or monsoon depressions. Cluster 11 (4.6% frequency) shows a classic monsoon depression pattern with very heavy rainfall (>15 mm/day) centered over the head of the Bay of Bengal and Bangladesh, extending into eastern India (Odisha, West Bengal). Cluster 4 (8.8%) seems to depict the later stage of a monsoon depression that has moved inland; and shows a rainfall maximum over central India (Madhya Pradesh region) and secondary maxima in the Western Ghats and NE India. The 850 hPa winds for Cluster 11 (Figure 5, last panel) show a pronounced cyclonic anomaly over the northern Bay of Bengal and eastern India, while Cluster 4 shows this anomaly shifted to central India, consistent with west-northwestward propagation of the system.
  - Cluster 9 (3.9%) is somewhat intermediate, it shows enhanced rainfall over northwest India (Rajasthan and the Kutch region) and adjoining Pakistan, as well as along the Himalayas. This suggests Cluster 9 could represent the remnants of a depression or a mid-tropospheric cyclone that has reached northwest India, bringing unusual rainfall to the desert region. The composite for Cluster 9 indeed shows a cyclonic circulation anomaly over Rajasthan (Figure 5) and increased precipitable water in that normally dry region (Figure 6). These three clusters collectively capture different stages and tracks of synoptic systems, from genesis in the Bay (Cluster 11) to mature phase over central India (Cluster 4) to decay or interaction with mid-level vortices in the northwest (Cluster 9). Their temporal succession (discussed in Section 3.3) often follows the life cycle of a monsoon depression.
  - 4. Widespread rain, active phase (WR-A) [1, 6, 7]: The remaining three clusters depict active monsoon conditions characterized by widespread rainfall not necessary tied to a monsoon depression. Cluster 7 (10.0% frequency) has a broad region of moderate to heavy rainfall covering central India, the west coast, and extending into the IGP, with a maximum around central India. It likely corresponds to a typical active monsoon spell without a strong depression influence, possibly driven by an active monsoon trough and embedded smaller-scale systems. Its wind anomaly (Figure 5) shows a weak cyclonic circulation spanning much of India and strong westerly anomalies over the Arabian Sea feeding moisture inland.
  - Cluster 1 (10.2%) shows an intense and extremely widespread rainfall pattern, heavy rain along the Western Ghats and west-central India, and significant rain even in otherwise drier regions. This cluster appears to represent the peak of an active monsoon when the monsoon trough is optimally positioned south of its mean and multiple systems (off-shore vortex, depressions, strong monsoon jet) act together. The composite winds show deep westerlies and low-pressure anomalies covering virtually the entire subcontinent.
- Cluster 6 (7.0%) is somewhat similar to Cluster 1 but with its maximum shifted slightly westward (covering Rajasthan and the Western Ghats simultaneously). Interestingly, Cluster 6 brings notable rainfall into the Thar desert region (similar to Cluster 9 but with also strong west coast rains), its occurrence suggests episodes when a mid-level cyclone over the Arabian Sea or an unusual westward extension of the monsoon trough brings moisture into northwest India. Clusters 1,

**Table 1.** Transition Probabilities Table for Indian Monsoon Rainfall Clusters: Emphasizing significant transitions with bold ( $\geq 0.1$ ) notations. may do it before and after.

|    | 1     | 2     | 3     | 4     | 5     | 6     | 7     | 8     | 9     | 10    | 11    |
|----|-------|-------|-------|-------|-------|-------|-------|-------|-------|-------|-------|
| 1  | 0.352 | 0.008 | 0.008 | 0.178 | 0.004 | 0.273 | 0.072 | 0.011 | 0.061 | 0.004 | 0.030 |
| 2  | 0.004 | 0.346 | 0.105 | 0.004 | 0.237 | 0.013 | 0.066 | 0.110 | 0.004 | 0.101 | 0.009 |
| 3  | 0.004 | 0.016 | 0.746 | 0.004 | 0.078 | 0.002 | 0.069 | 0.034 | 0.001 | 0.006 | 0.042 |
| 4  | 0.024 | 0.019 | 0.069 | 0.379 | 0.024 | 0.045 | 0.082 | 0.103 | 0.210 | 0.008 | 0.037 |
| 5  | 0.001 | 0.033 | 0.220 | 0.005 | 0.424 | 0.002 | 0.098 | 0.106 | 0.002 | 0.080 | 0.028 |
| 6  | 0.066 | 0.019 | 0.002 | 0.058 | 0.015 | 0.514 | 0.139 | 0.026 | 0.109 | 0.019 | 0.032 |
| 7  | 0.064 | 0.023 | 0.067 | 0.026 | 0.056 | 0.086 | 0.550 | 0.040 | 0.019 | 0.019 | 0.050 |
| 8  | 0.017 | 0.027 | 0.131 | 0.030 | 0.090 | 0.008 | 0.059 | 0.516 | 0.004 | 0.023 | 0.094 |
| 9  | 0.051 | 0.007 | 0.118 | 0.010 | 0.068 | 0.057 | 0.108 | 0.064 | 0.389 | 0.007 | 0.122 |
| 10 | 0.000 | 0.057 | 0.081 | 0.000 | 0.254 | 0.018 | 0.046 | 0.088 | 0.007 | 0.438 | 0.011 |
| 11 | 0.050 | 0.015 | 0.054 | 0.173 | 0.037 | 0.024 | 0.045 | 0.158 | 0.011 | 0.011 | 0.423 |

6, and 7 represent different intensities and extents of active conditions, from moderate active spells (Cluster 7) to extreme widespread rainfall (Cluster 1).

## 3.3 Cluster Transition Dynamics





Transitions from one cluster to another on day-to-day basis with probability  $P_{ij} > 0.1$  are shown in the Figure 7, while transition probabilities of all clusters are provided in Table 1. A high  $P_{ii}$  on the diagonal indicates day-to-day persistence of the clusters, whereas off-diagonal high values indicate transitions. The most prominent feature is the strong persistence (high self-transition probability  $P_{ii}$ ) of Cluster 3, the prolonged break monsoon state. Table 1 shows  $P_{33} \approx 0.746$ , meaning that nearly 75% of the time, a break-monsoon day is followed by another break-monsoon day. This aligns with the well-known tendency for breaks to last several days to weeks (Raghavan, 1973). In contrast, the active monsoon clusters with monsoon depression (Clusters 4, 9, 11), are less persistence with  $P_{ii} \approx 0.4$ , indicating that active conditions tend to continue, but with less persistence than the breaks. Cluster 1 (extremely widespread rain) has a notably lower persistence ( $P_{11} \approx 0.35$ ), which is reasonable since such intense widespread rainfall is usually a transient peak of an active spell and often transitions to a slightly weaker state afterward (e.g., Cluster 6 or 7). The high persistence of Cluster 6 ( $P_{66} \approx 0.514$ ) is interesting because Cluster 6 represents an active state focused on the western India (including Thar/Kutch and West Coast), which might be linked to mid-level vortex situations that can linger quasi-stationarily over the Arabian Sea/western India, producing several consecutive days of rainfall there.

Break clusters (2, 5, 8, 10) generally have a tendency to transition to other break-type clusters or to the prolonged break Cluster 3. For example, Cluster 2 and 10 (both NE-focused break) frequently transitions to Cluster 5 (another NE-Break clusters) (with high probability around 0.24) and Cluster 2 also has transition probability of > 0.1 % to go to Cluster 3, 8 and






10 (see Table 1). Similarly, Cluster 8 (which has rainfall in central India but with break-like circulation) often transitions to Cluster 3 (with  $P_{83} = 0.131$ ). These indicate that once the monsoon enters a weak phase, it often oscillates among the break-type patterns (with rainfall shifting between NE India and perhaps the foothills) before eventually either recovering to an active state or intensifying to a different pattern. Conversely, transitions from break clusters directly to strong active clusters are rare (typically P 





to break (Cluster 3) are low probability (around 0.002 to 0.069), meaning active spells usually decay gradually (perhaps via cluster 8 or via a depression moving out).

Within the break clusters, there is a fair amount of transitioning except for Cluster 3 (prolong break). For example, Cluster 2 (NE heavy rain break) can go to Cluster 5 or 10 (other NE rain variants) with notable probabilities, as well as to Cluster 8 (which has some central India rain) with  $P_{2,8} = 0.110$ . Cluster 5 (NE focus) transitions to Cluster 8 and 10 quite often. So, even during an overall break spell (with trough at foothills), the exact distribution of rainfall can oscillate among NE India, the Himalayas, and maybe a weak low in central India that briefly brings some rain there (Cluster 8). These transitions might correspond to Interaseasonal oscillation, possibly related to westward propagating sub-seasonal modes (such as Suhas et al., 2013).

## 3.4 Seasonality of Clusters and Long-Term Changes

Figure 8 shows monthly (June–September) rainfall contributions of each cluster for Thar & Kutch (T&K), Indo-Gangetic Plains (IGP), and Northeast India (NEI) regions as shown in the Figure 3. In T&K, most seasonal rainfall occurs in July–August, dominated by Cluster 6 and 9; Cluster 6 alone exceeds 30%. June and September have higher shares of Cluster 3 (break phase), consistent with shorter active spells during onset and withdrawal. In the IGP, contributions are more evenly distributed among active clusters, with Cluster 1 and 7 prominent in July–August; Clusters 2 and 8 contribute moderate foothill-related rainfall even during break-like phases. NEI receives substantial break-phase rainfall, with Clusters 2, 5, and 10 contributing large early-season fractions; active clusters (1, 4, 7) add mid-season rainfall, but break clusters remain important.

Figure 9 presents JJAS cluster-frequency trends for 1961–2018. Cluster 3 (break conditions) shows a slight but statistically non-significant upward tendency. Clusters 2 and 5 (NEI rainfall during breaks) display a pronounced decline, with Cluster 2 virtually absent after 1975, consistent with the observed Northeast India drying. Cluster 8 (intermediate rainfall) remains relatively stable, whereas Clusters 4 and 9 (monsoon depressions) show modest increases after 2000. Cluster 6 is stable to slightly rising in recent years. Cluster 1 shows an increasing trend before 1980 but stabilizes thereafter. Cluster 9 is too infrequent for robust detection though it may have increased.

Figure 10 quantifies regional rainfall changes between 1961–1989 (R1) and 1990–2018 (R2) in terms of intensity and frequency contributions and links them with the long-term cluster trends in Fig. 9. In T&K, the  $\sim$ +30 mm (+15 %) increase arises mainly from Cluster 9 frequency gains (+20 mm) together with slight intensity increases, and a  $\sim$ +10 mm contribution from Cluster 6 intensity rise. This is consistent with the gradual post-1990 increase of Cluster 6 and 9 frequencies visible in Fig. 9. Over the IGP, rainfall decreases by  $\sim$ 50 mm (-7 %), driven equally by reduction in frequency and intensity of Cluster 8 (-25 mm) and Cluster 2 frequency reduction (-15 mm). These declines match the downward trend of Cluster 2 frequency and the relatively stagnant or declining Cluster 8 frequency in Fig. 9. However, increased occurrences of Clusters 1, 5, 9 and 10 are adding rainfall to the annual total. CMIP5 and CMIP6 projections similarly indicate increasing monsoon rainfall over this region, a signal already evident in the observed cluster changes (Kulkarni et al., 2020).

In NEI, the  $\sim$ 150 mm (-9%) rainfall decline is driven by the largest reduction in Cluster 2 frequency. Although Clusters 5, 9 and 10 show frequency increases comparable in magnitude to the Cluster 2 loss, the cluster 5 has increasing trend in







the frequency before 1990 and declining sionce then wehich this decomposition analysis (Eq. 3) is not able to capturing. In addition clusters 9 and 10 contribution is offset intensity reductions across Clusters 7–11 (7, 8, 9, 10 and 11), yielding a net deficit. This highlights a shift in NEI focused rainfall during Cluster 2 and 5 break-season mode to other mix clusters and monsoon depressions.

## 4 Conclusions and Remarks

This study presents a diagnostic framework linking unsupervised rainfall clustering to synoptic-scale drivers and long-term trends of the Indian Summer Monsoon (ISM). By combining IMD high-resolution rainfall data with NCEP–NCAR reanalysis fields, eleven distinct rainfall regimes were objectively identified and associated with low-level wind, sea-level pressure, and moisture anomalies to reveal clear separation between active and break phases, quantifie regime transitions, and attribute regional rainfall trends to changes in frequency and intensity of specific clusters.

Our results show strong spatial heterogeneity in ISM variability and trends. Over Thar & Kutch (T&K), increased rainfall since 1990 is primarily linked to higher frequency of mid-tropospheric cyclones and westward-propagating systems (Clusters 6 and 9), which transport moisture into arid zones. In contrast, Indo-Gangetic Plains (IGP) and Northeast India (NEI) experienced significant drying, driven by decreased frequency and weaker intensity of break-phase clusters embedded within the monsoon trough (especially Cluster 2) and more persistent dry-break regimes (Cluster 3). The cluster transition analysis confirms that break spells are highly persistent while monsoon depressions are more transient, evolving systematically from formation over the Bay (Cluster 11) to inland propagation (Cluster 4) and decay over northwest India (Cluster 9). These dynamics closely mirror observed synoptic life cycles and show that active spells typically decay gradually via intermediate states rather than abrupt shifts to break conditions.

Seasonality analysis highlights that T&K rainfall peaks in July–August due to active-phase clusters, while NEI rainfall has historically been dominated by break-phase clusters, which have declined sharply after the late 1970s. The decomposition of rainfall change between 1961–1989 and 1990–2018 shows that NEI drying ( $\sim$ 150 mm or -9%) stems from loss of Cluster 2 frequency and reduced intensities across several active clusters, whereas IGP drying ( $\sim$ 50 mm or -7%) reflects fewer intermediate active regimes and weaker break-phase rainfall. Conversely, T&K rainfall increase ( $\sim$ +15%) arises from more frequent northwest-focused active systems. These findings suggest that regional monsoon rainfall changes cannot be explained solely by large-scale weakening of the mean monsoon flow (Turner and Annamalai, 2012), but rather by the modulation of synoptic regimes and their transitions.

This study demonstrates that regime-based analysis provides a powerful diagnostic tool to evaluate model simulations, attribute observed rainfall changes, and understand intraseasonal-to-decadal variability. Future work should focus on high-resolution modeling to resolve trough displacement, vortex dynamics, and orographic—flow interactions that govern cluster evolution. The presented cluster dataset (Raut, 2025) offers a benchmark for investigating teleconnections (ENSO, IOD, MJO) and assessing model biases in reproducing ISM rainfall regimes and their transitions.

https://doi.org/10.5194/egusphere-2025-4585 Preprint. Discussion started: 6 November 2025

Data availability. India Meteorological Department (IMD) binary rainfall files and Grid Analysis and Display System (GrADS) control files were acquired from the India Meteorological Department, Pune, available at https://www.imdpune.gov.in/cmpg/Griddata/Rainfall\_25\_ 395 Bin.html. National Centers for Environmental Prediction (NCEP) and National Center for Atmospheric Research (NCAR) reanalysis-1 data were obtained from the Physical Sciences Laboratory of the National Oceanic and Atmospheric Administration (NOAA), accessible at https://psl.noaa.gov/data/gridded/data.ncep.reanalysis.html.

Code and data availability. The code utilized in the analysis and plotting can be accessed at https://github.com/RBhupi/monsoon\_clusters 400 The cluster data produced in this study are available at (Raut, 2025).

Author contributions. BR and AD contributed to the writing and editing of the paper. BR provided the original idea, leading the methodology design, conducting sensitivity tests, statistical analysis, and plotting. AD was responsible for analyzing synoptic data and plotting key figures. SW, SI and DK handled data procurement and curation. DK and PN conducted initial analysis under the guidance of AD and BR. PS and PK provided general advice and guidance.

Competing interests. The authors declare no conflict of interest.

Disclaimer. This work was not part of any institutional or externally sponsored project, and was completed entirely in authors' personal time.

Acknowledgements. The authors gratefully acknowledge the India Meteorological Department (IMD), Pune, for providing high-resolution daily rainfall data, and the NCEP-NCAR for the Reanalysis data.

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

**Figure 4.** Mean spatial patterns of eleven clusters of gridded rainfall **20**mputed over Indian landmass using the methodology described in the section 2.4 and their percent frequency of occurrence.

**Figure 5.** Composites of mean sea pressure anomalies (shaded) overlaid with 850 mb wind anomalies (vectors) for 11 clusters over larger region using the methodology described in the section 2.

Figure 6. Composites of geopotential height anomalies (shaded) (850mb) overlaid with precipitable water (contours) for each of the clusters.

Figure 7. Transition of rainfall clusters shown with directed arrows weighted by probabilities for  $P_t > 0.1$ . Table 1 shows all transitions. The Transition Probabilities were computed according to section 2.5.

**Figure 8.** The monthly contributions of each cluster (stacked color strips) and total (total height of the bars) are shown as percentages of seasonal rainfall in K&T, IGP, and NEI regions.

**Figure 9.** Seasonal frequency of occurrence of clusters from 1961 to 2018. The blue line shows smoothed data obtained using locally weighted smoothing (LOESS). Rapid decline in rainfall for NEI clusters 2 and 5 is evident.

**Figure 10.** Contribution of cluster frequency and intensity components to the total change in seasonal rainfall between 1961–1989 and 1990–2018 over the KT, IGP, and NEI regions, computed using Eq. 3. The smaller residual ('noise') term shows second-order variations, showing the significance of frequency and intensity terms in the total change.