# Peer review of "Rainfall regimes, their transitions, and long-term changes during Indian summer monsoon"

_EGUsphere, 2025_

## Referee Comment (RC2)

**Review Report**

Title: Rainfall regimes, their transitions, and long-term changes during Indian summer monsoon
Author(s): Bhupendra Raut et al. (2025)
MS No.: egusphere-2025-4585

**General comment:**

I find the motive of identifying clusters important. However, I find the link between the clusters and the regional long-term trends muddy and not well established by the existing arguments. Overall, the narration requires sharpening.

The manuscript first identifies seasonal mean rainfall changes over India over the duration 1961–2018 (Figure 3). It does so by splitting the whole duration into two periods, 1961–1989 (earlier period) and 1990–2018 (later period), and then subtracting the seasonal mean rainfall of the earlier period from that of the later period. By performing this analysis, the authors identify three regions that exhibit noticeable trends, namely Thar and Kutch (west north-western part of India), Indo-Gangetic plains, and the Northeast Indian region (in fact, the easternmost part of India). It is noteworthy that the reliability of the data the authors have used is debatable over the Northeast Indian region (Zahan et al., 2021).

Then the authors decompose JJAS rainfall over India into 11 clusters or spatial-patterns (Figures 4, 5, and 6) and analyze their transition probabilities (Figure 7). They group these 11 clusters into 4 groups. Is this done only based on the transition probabilities? The grouping requires a better argument and quantification.

Further, they compute monthly contributions of these clusters to the three regions (Figure 8) identified in Figure 3. It is not clear how this was computed. From Figure 8, it seems the authors computed the seasonal mean of each cluster and then computed its percentage relative to the total seasonal mean, averaged over each region indicated by the boxes in Figure 3 (please explain this in detail in the relevant section of the manuscript). The narration of Figure 8 is not transparent enough. There is considerable confusion regarding the interpretation of Figure 8.

As stated, Figure 8 indicates considerable contribution from specific clusters to specific regions. However, despite finding that specific clusters contribute to specific regions, the authors analyzed the "Seasonal frequency of occurrence of cluster" for all 11 clusters. Interestingly enough, especially because some clusters earlier were grouped, the clusters of the same group exhibit different trends in Figure 9. For example, clusters 2, 5, 8, and 10 in Figure 9. The authors also did not describe Figure 9 well. What is "Seasonal frequency of occurrence of cluster"? Is it $N_i$ in Equation 3?

In Figure 10, the last figure of the manuscript, the authors compute changes in rainfall corresponding to each cluster and decompose that change into intensity and frequency change

(following Catto et al., 2012; the authors should refer to relevant citations while discussing the results, in addition to mentioning them in the introduction or data-and-methodology section). Like Figure 9, the description of Figure 10 is also muddy. The green bar corresponding to cluster 9 for the T&K region (that is mentioned as "Kutch and the Thar" in Figure 10; please make it consistent with the rest of the manuscript) goes past 40 mm. Then why do you mention "Cluster 9 frequency gains (+20 mm)" in the manuscript? Why do you not discuss negative contributions from cluster 2 for T&K whereas you do discuss those from clusters 2 and 8 over IGP? For T&K, IGP, and NEI, the clusters that were found to contribute most in Figure 8 and Figure 10 are not exactly the same. How can we reconcile this?

After performing the above analysis, the authors claim that:
- This study presents a diagnostic framework linking rainfall clusters to synoptic-scale drivers: I fail to see any mechanism or statistics in the analyses presented in the manuscript supporting this claim.
- This study identified eleven distinct rainfall regimes: I also fail to understand how 11 distinct regimes were identified if the authors argue that some of the clusters are actually dynamically similar and can be put under one group, forming a total of 4 groups.

Based on the above reasoning, I recommend that the manuscript requires major revision.

**Detail comments:**
1) Title: The title can be more conclusive.
2) Abstract: Unclear. Also, the authors ambiguously use the words "intensity" and "frequency" in the abstract. It is not clear if they mean these for rainfall or for clusters.
3) Introduction: It reads pedagogically rather than as an introduction to a research manuscript. Evidence of this is the consistent use of 50-year-old references. Paragraph #1 introduces the monsoon in general. The subsequent paragraphs introduce land–ocean contrast, the ITCZ, monsoon evolution, monsoon trough, depressions, mid-tropospheric cyclones, orographic effects, intraseasonal variability, and teleconnections. Then, in the paragraph starting at line #76, the authors very quickly mention a large number of features of the monsoon. In the paragraph at line #87, the authors direct attention to extremes: past evidence of their increase over India (central India), arguments for their projected continued increase (Clausius–Clapeyron logic), and finally discuss future projections of increasing extremes. In the final paragraph, the authors introduce the aims of this study. The three aims, nice and interesting as they are, are not related to the previous paragraphs of the introduction. The three aims mention links between rainfall and atmospheric circulation, their evolution, and their impact on monsoon rainfall trends. They also mention a future scope relevant to teleconnections and model biases. The bottom line is that the introduction is not sharp enough, not updated enough with relevant and recent references, and does not provide adequate scientific background..
4) Data and methodology: The section is fairly well written. I have only two concerns:

a) Why not use ERA5 data?
b) Are the results consistent with Zahan et al. (2021) over NEI? It seems consistent. Nonetheless, please comment.

[Zahan, Y., Mahanta, R., Rajesh, P. V., and Goswami, B. N. (2021). Impact of climate change on North-East India (NEI) summer monsoon rainfall. Climatic Change, 164, 2. https://doi.org/10.1007/s10584-021-02994-5]

5) Results:

3.1:

Finds rainfall trends:

T&K - increase : consistent with poleward migration of monsoon winds and more rain over desert regions

Indo-Gangetic plains - decrease : reported earlier in some studies.

NE India - decrease

These are already reported for observations. Please cite relevant studies.

3.2 Grouping of clusters is debatable

Northeast rainfall, break phase (NE-B) [2, 5, 8, 10]: I agree 5 and 10 look similar. 2 seems to be somewhat different. Especially focussing on the low-level winds (Figure 5). Cluster 8 definitely looks different. Grouping of clusters will always remain debatable if it is based on visual inspection unless the authors can argue based on some matrix that can quantify the degree of association of clusters.

Monsoon depressions, active phase (MD-A) [4, 11, 9]: 11 is over the T&K region and 9 is NE region. Are they the same? Your rainfall trend analysis says they are not.

3.3 Transition probabilities discussed are agreeable. Noteworthy that authors mention "Cluster 8" acts as an independent category. It goes back to my comment on the debatable logic behind grouping of clusters.

Another comment on the transition probabilities. Table 1 can be converted to a heat-map with warmer colors for higher values and cooler colors for smaller values to make it visually more communicative. I mean, the table would remain a table but each cell will have a color. The nonsignificant transition probabilities may be omitted or not colored in the heat-map. Also, Figure 7 can be omitted (or moved to supplementary).

3.4 (paragraph centered around L#340): These observations are consistent with the cluster spatial patterns. But at the same time, aren't these statements redundant since from the cluster spatial patterns (Figure 4) it is obvious that rainfall contributions over T&K, IGP, and NE would dominantly come from (6,9), (1,7), and (2,5,8,10) respectively?

On a closer observation, I notice from Figure 8 that, IGP has a lot of contributions from Cluster-8 and also cluster-6 (in July) and I fail to see contributions from Cluster-1. Also, for NE I see a lot of contributions from Cluster-3. Maybe I am not

reading the plot well. In my opinion, Pie Diagrams or Stacked Violin Plots might be a better option instead of a stacked bar plot. In any case, the narration requires a lot more transparency.

L#347: "consistent with the observed Northeast India drying" : Please provide either evidence or reference.

6) Conclusion and remarks: Claims are unsupported by analyses presented. For example, "Our results show strong spatial heterogeneity in ISM variability and trends. Over Thar & Kutch (T&K), increased rainfall since 1990 is primarily linked to higher frequency of mid-tropospheric cyclones and westward-propagating systems (Clusters 6 and 9), which transport moisture into arid zones." Which analysis evidentially supports this claim? I only see your statement, "Another cluster that shows interesting transition behavior is Cluster 9 (northwest-focused active). It has a significant probability to transition to Cluster 3 (P9,3 = 0.118), meaning after a rain event in the northwest (often due to a mid-level cyclone or dying depression), the monsoon likely goes into a break"